# Regional Adaptability of Global and Regional Hydrological Forecast System

**Han Wang** [1], **Ping-an Zhong** [1,2,*], **Ervin Zsoter** [3,4], **Christel Prudhomme** [3,5,6], **Florian Pappenberger** [3,7] and **Bin Xu** [1,2]

1 College of Hydrology and Water Resources, Hohai University, No. 1 Xikang Road, Nanjing 210098, China
2 National Engineering Research Center of Water Resources Efficient Utilization and Engineering Safety, Hohai University, No. 1 Xikang Road, Nanjing 210098, China
3 European Centre for Medium-Range Weather Forecasts (ECMWF), Reading RG2 9AX, UK
4 Department of Geography and Environmental Science, University of Reading, Reading RG6 6AH, UK
5 UK Centre for Ecology and Hydrology (UKCEH), Wallingford OX10 8BB, UK
6 Department of Geography and Environment, University of Loughborough, Loughborough LE11 3TU, UK
7 Department of Civil, Chemical, Environmental and Material Engineering, Università di Bologna, 0136 Bologna, Italy
* Correspondence: zpa_hhu@163.com or pazhong@hhu.edu.cn; Tel.: +86-135-0518-5185

**Abstract:** Our paper aims to improve flood forecasting by establishing whether a global hydrological forecast system could be used as an alternative to a regional system, or whether it could provide additional information. This paper was based on the operational Global Flood Awareness System (GloFAS) of the European Commission Copernicus Emergency Management Service, as well as on a regional hydrological forecast system named RHFS, which was created with observations recorded in the Wangjiaba river basin in China. We compared the discharge simulations of the two systems, and tested the influence of input. Then the discharge ensemble forecasts were evaluated for lead times of 1–7 d, and the impact on the forecasts of errors in initialization and modelling were considered. We also used quantile mapping (QM) to post-process the discharge simulations and forecasts. The results showed: (1) GloFAS (KGE of 0.54) had a worse discharge simulation than RHFS (KGE of 0.88), mainly because of the poor quality of the input; (2) the average forecast skill of GloFAS (CRPSS about 0.2) was inferior to that of RHFS (CRPSS about 0.6), because of the errors in the initialization and the model, however, GloFAS had a higher forecast quality than RHFS at high flow with longer lead times; (3) QM performed well at eliminating errors in input, the model, and the initialization.

**Keywords:** discharge ensemble forecast; global hydrological forecast system; regional hydrological forecast system; influence factors

## 1. Introduction

Floods and droughts have been reported as the costliest and most destructive natural disasters in China. The resulting direct economic losses are projected to rise even further in the coming decades, posing a serious threat to the safety of people's lives and property and the stable development of the social economy [1]. Hydrological forecasts are a prerequisite for disaster relief by providing timely information on where and when floods and droughts will occur in the near future. In addition, reliable hydrological forecasts are necessary to prepare for an appropriate response in the management of hydropower reservoirs, water supply, agriculture, and navigation [2].

Traditionally, hydrological variables have been predicted by forcing hydrological forecast systems with observed meteorological data to estimate the forthcoming hydrological conditions, and the lead time of the resulting hydrological forecast approximately equals the concentration time. By contrast, hydrological ensemble forecast systems generate probabilistic forecasts and corresponding uncertainty information by forcing a hydrological

model with ensemble weather forecast data, and have gained extensive attention and application because of their clear advantages [2–6]. Moreover, compared with traditional hydrological forecasts, hydrological ensemble forecasts can effectively prolong the lead time [7] and generate multiple forecast results for the same moment in time and location [8,9]. High-quality hydrological ensemble forecasts can provide disaster prevention and control departments, and water resource management departments, with confidence in their decision-making [2].

Hydrological forecast systems can be divided, according to the size of the area they cover, into global hydrological forecast systems and regional/local hydrological forecast systems. A regional/local hydrological forecast system is designed for a specific location or river basin. It is based on model parameter calibration using regional/local observations, and is the tool most commonly used by researchers and commercial industries for a wide range of projects and for developing value-added products in the area, such as evaluating discharge forecasts from different perspectives by using a regional forecast system [10–13], developing methods to improve forecast skill [11,13–17] and studying the uncertainty in hydrological ensemble forecasting [15,16,18]. Setting up a global hydrological forecast system is computationally demanding but could be valuable, especially for developing regions of the world, where effective regional hydrological forecast systems are scarce. For example, Passerotti et al. [19] studied a flood early warning system using the prior notification of the Global Flood Awareness System (GloFAS) in the Sirba River; Bischiniotis et al. [20] assessed the skills of GloFAS in Peru for the years 2009–2015, which does not have its own flood forecasting system; the forecasts from GloFAS were used in the Uganda Red Cross Society forecast-based financing pilot project, to ensure that automatic, prefunded early action would be triggered by forecasts in this data-scarce location [21]. A global system would also provide additional information to basin management departments, even if there already was a mature regional hydrological forecast system [3].

The Global Flood Awareness System (GloFAS) [22] is one of the best-known global hydrological forecast systems and is the global flood service of the European Commission's Copernicus Emergency Management Service (CEMS). It was set up by the European Centre for Medium-Range Weather Forecasts (ECMWF), in collaboration with the Joint Research Centre (JRC) of the European Commission and the University of Reading. The system became preoperational in July 2011 and fully operational as a 24/7 supported service in April 2018. It is freely available through an open license and is designed for decision makers and forecasters in national and international water authorities, bodies responsible for the management of water resources, hydropower companies, civil protection authorities, and international humanitarian aid organizations, as reflected by the more than 5000 registered users [22]. Most studies evaluating GloFAS take the observed discharge as the evaluation standard. For example, Alfieri et al. [23] evaluated GloFAS's performance against observed discharge from 620 stations and a qualitative analysis of a case study on a Pakistan flood. Alfieri et al. [24] also evaluated a GloFAS discharge simulation against a global network of 1801 stations providing daily river discharge observations, while Bischiniotis et al. [20] compared a simulated GloFAS discharge against observations for 10 river gauges. In addition, Senent et al. [25] assessed the potential of GloFAS for calibrating hydrological models in ungauged basins, and found it had substantial potential for calibrating hydrological models in monthly streamflow. However, there is also a need for a comparative study of GloFAS and regional hydrological forecast systems, comparing not just the simulations against observations, but also evaluating whether the application of GloFAS in areas without observed data could be a viable alternative.

When running hydrological forecast systems in simulation mode (i.e., by forcing the systems with (proxy-) observed weather variables) [14], the consequent simulation can be used to evaluate how well the system is adapted to the basin, as the error of the simulation is expected to be driven by the model error resulting from the imperfect model structure and inaccurate parameterization. However, it is a challenge to access observations of sufficient quality and length for a global hydrological forecast system [3], so in addition to

the model error, the input error of such a system will also influence the simulation. As for the forecasting mode, hydrological forecast errors result from various interactive sources of error, of which the most common relate to meteorological input, modelling, and initial conditions [26]. The meteorological forcing errors are due to uncertainty in the numerical weather prediction models. In this context, it is necessary to consider the model errors and the input errors when evaluating discharge simulations and verifying forecasts.

Our primary goal with this study was to examine whether a global hydrological forecast system could be a good alternative to a regional hydrological forecast system, and to answer the following two main questions: Are global hydrological forecast systems always inferior to regional hydrological forecast systems? Which factors affect the performance of forecast systems? To this end, we first analyzed the discharge simulations of two prediction systems at different scales in order to understand the influence of input data and the improvement generated by post-processing. Then, we estimated the quality of the discharge forecasts produced by the two systems, focusing on the contributions of initial error and model error, as well as post-processing. The flow chart of the manuscript is shown in Figure 1.

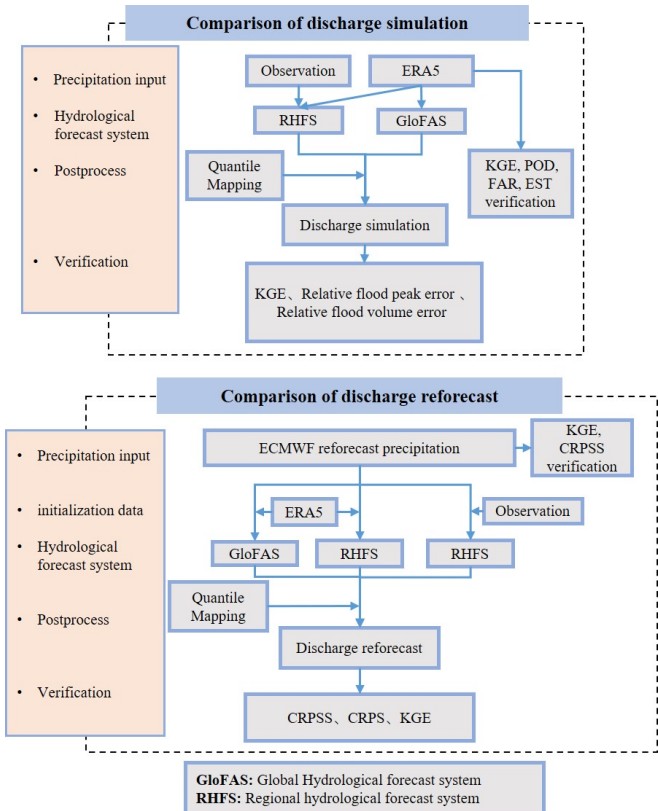

**Figure 1.** Flow chart of this paper.

The remainder of this paper is organized as follows: The materials and methods including study area, data, hydrological forecast systems, experimental design, post-processing methods and verification metrics are described in Section 2. The results are presented in Section 3, followed by the discussion and conclusion in Section 4.

## 2. Materials and Methods

### 2.1. Study Area

The Wangjiaba basin is the upstream watershed of the Huai River in China, with a river length of 360 km, a drop of 178 m and a basin area of 30,630 km$^2$. The average annual precipitation over the selected basin is about 1000 mm, with great inter-annual variability

and uneven temporal and spatial distributions. The flood season of the basin is from June to October, with 60% of the annual precipitation concentrating between June and August and most rainstorms occurring in June and July, producing river discharge peaks between June and October. Figure 2 shows a map of the watershed with the observing stations and local topography.

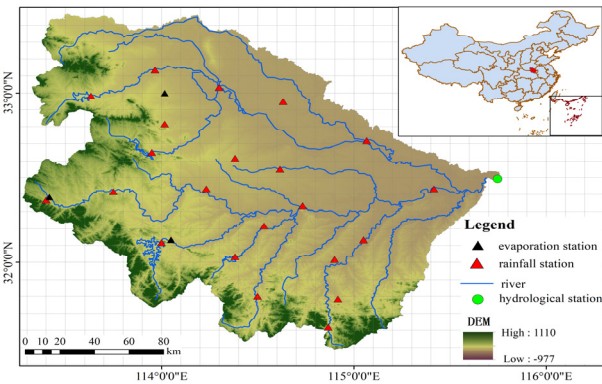

**Figure 2.** Map of the study area showing the terrain elevation, main rivers of the Wangjiaba basin, the location of the selected hydrological station, rainfall stations and evaporation stations. The gray grids represent the 0.1 × 0.1 degree resolution of GloFAS. The inset map shows the approximate location of the study area in China.

### 2.2. Dataset

The dataset used in this paper included observed data and ERA5 data, as well as data from the GloFAS reanalysis, the ECMWF reforecast and the GloFAS reforecast.

The observed data for the study catchment contained daily pan evaporation, observed daily discharge, reservoir water levels and daily precipitation. Daily pan evaporation data were drawn from the Chinese National Meteorological Information Center, and the rest were drawn from the Annual Hydrological Report P.R. China, published by the Hydrological Bureau of the PRC Ministry of Water Resources. The observed discharge data were converted into natural discharge using the water balance method, in advance.

ERA5 is a global atmospheric reanalysis product of ECMWF. The complete ERA5 data covers the period from 1950 to near real time [27]. It is available from the Copernicus Climate Change Service (C3S) Copernicus Climate Data Store (CDS) with a regular latitude/longitude grid. For the regional hydrological forecast system, we used ERA5 daily precipitation data at a resolution of $0.28° \times 0.32°$.

The GloFAS reanalysis is a time series consisting of simulated daily data of river discharge from the GloFAS system produced by CEMS. It is generated by forcing the GloFAS operational modelling chain with meteorological variables from ERA5 to provide data for over 40 years from the recent past, from 1979 to the present [22]. The dataset includes the variables of river discharge and the upstream area for each GloFAS grid cell with a resolution of $0.1° \times 0.1°$. The data are freely available through the CDS at https://cds.climate.copernicus.eu/#!/home (accessed on 10 January 2022).

We chose the ensemble weather forecasts of ECMWF as the forcing data in the hydrological forecast systems because of their proven good performance in China [28–31]. This study used reforecast data, which are forecasts of past dates reproduced with a global forecast system as close to the operational system as possible. ECMWF reforecasts are global-scale forecast runs that use the same integrated forecasting system (IFS) model version as the real-time ensemble forecasts (ECMWF-ENS), using data from the past 20 years, with lead times of up to 46 days and 11 forecast members. The data are initialized twice weekly (Monday and Thursday), and have a horizontal resolution of 18 km for a lead time of up to 15 days, and of 36 km for longer forecast lead times of up to 46 days. Reforecasts are available through the ECMWF's Meteorological Archival and Retrieval System (MARS; https://apps.ecmwf.int/mars-catalogue/ (accessed on10 January 2022)). In this paper, we

used ECMWF reforecasts produced for the calendar year of 2019, using cycle 45R1 before 11 June 2019, and cycle 46R1 after that date. The regional model and all meteorological forcing for GloFAS used precipitation reforecasts with a resolution of 0.125°.

The GloFAS reforecast is a simulated time series of river discharge, produced by forcing the hydrological modelling chain with ECMWF reforecasts. The initial conditions were provided by the GloFAS reforecast for the corresponding date [22]. GloFAS reforecast datasets provide daily reforecasts of river discharge across the world for up to 46 days with twice weekly start dates, for 20 years in the recent past. The GloFAS reforecast has a resolution of $0.1° \times 0.1°$ with 11 ensemble members and is freely available to download through the CDS.

### 2.3. Methods

#### 2.3.1. Hydrological Forecast Systems

The regional hydrological forecast system and global hydrological forecast system used in this paper are introduced below. The two systems differ in the applied hydrological model, calibration method and calibration data.

#### Regional Hydrological Forecast System

This paper used a regional hydrological forecast system named RHFS. The Xinanjiang model was used as the hydrological model of RHFS, and the system was calibrated using meteorological observations and natural discharge, based on the particle swarm optimization algorithm.

The Xinanjiang model is a conceptual lump model developed in 1973 and described in an international publication [32]. The model is widely used in humid and sub-humid regions in China. The inputs of the model are basin–mean precipitation and basin–mean pan evaporation, while the outputs are basin outlet discharge and basin evaporation. A detailed description of the Xinanjiang model can be found in Appendix B.

A particle swarm optimization algorithm (PSO) is a parallel mechanism of computational intelligence algorithm, based on continuous searching for the random initialization particles (each feasible solution of the problem is called a particle) in the solution space according to the direction and distance indication, to achieve the optimal result [33]. PSO is widely used in various fields due to it being easy to understand and implement, and its strong ability of local optimization [33].

#### Global Hydrological Forecast System

This article uses datasets from GloFAS version 3.1, released on 2021-05-26, which incorporates the latest version of hydrological datasets. GloFAS v3.1 is based on the LISFLOOD hydrological model, which is a spatially distributed semi-physical and grid-based model [34], calibrated against observed daily streamflow for 1226 river basins with a total watershed area of 51 million km$^2$ (the catchment studied in this paper is not included) across the globe, using ERA5 meteorological data as forcing data, and the evolutionary algorithm DEAP as the parameter optimization method [24].

#### 2.3.2. Modeled Design

When selecting the time period used in this study, we aimed at including as many floods as possible and, therefore, chose 6 flood seasons (from June to October) containing relatively large floods, namely, the flood seasons of 2007, 2008, 2010, 2015, 2016 and 2017. The flood seasons of 2010 and 2015–2017 were used for calibration and were called the calibration period, while the flood seasons of 2007 and 2008 were used for verification and were called the verification period. All data used in this study were on a daily time step, and the resolution of the GloFAS data were $0.1° \times 0.1°$. ECMWF precipitation reforecasts had a resolution of $0.125° \times 0.125°$, and ERA5 precipitation data had a resolution of $0.28° \times 0.32°$. Additionally, ECMWF precipitation reforecasts and GloFAS river discharge reforecasts were collected for lead times of 1-7 d with 11 ensemble members.

The calibration of RHFS was carried out for the calibration period (2010, 2015–2017, June–October of each year), using basin–mean precipitation and pan evaporation input, estimated by inverse distance weighting (IDW) from gauged measurements. For every PSO run, the particle size and the number of iterations were both set to 100, with the objective function minimizing the root mean square error (RMSE) between the simulated discharge and the corresponding natural discharge, so that the number of objective function evaluations was $100 \times 100 = 10{,}000$. After calibrating RHFS by PSO, we manually relaxed the boundaries if the optimal parameters fell on their boundaries (i.e., the boundaries were preset not high or not low enough), and then repeated the PSO run with the modified boundaries until all optimal parameters fell within them.

We used the regional hydrological forecast system (with RHFS) and the global hydrological forecast system (with GloFAS) to generate discharge simulations and forecasts. Simulations and forecasts used in this paper are briefly introduced in Table 1 and described in more detail in Simulation and Forecast part.

**Table 1.** Name, system, input, initialization data and post-processing of simulations and forecasts used in this study; the versions that use quantile mapping are in parentheses.

| Mode | Name | System | Input | Initialization Data | Post-Processing |
|---|---|---|---|---|---|
| Simulations | RHFS-S (RHFS-S-QM) | RHFS | Observation | | No (yes) |
| | GloFAS-S (GloFAS-S-QM) | GloFAS | ERA5 | | |
| | RHFS-S-ERA5 (RHFS-S-ERA5-QM) | RHFS | ERA5 | | |
| Forecasts | RHFS-F (RHFS-F-QM) | RHFS | ECMWF reforecast | Observation | |
| | GloFAS-F (GloFAS-F-QM) | GloFAS | | ERA5 | |
| | RHFS-F-ERA5 (RHFS-F-ERA5-QM) | RHFS | | ERA5 | |

Simulation

In order to examine the influence of input data and post-processing (Table 1), we ultimately analyzed six different simulations:

1. RHFS-S was the simulation of the RHFS forced with meteorological observations, run separately in each year from 1 June to 31 October;
2. GloFAS-S is short for GloFAS reanalysis, and was the reanalysis simulation generated by the global hydrological forecast system GloFAS, forced with ERA5 as proxy-observations. We used the pixel that best represented the study basin in the GloFAS river network at a resolution of $0.1° \times 0.1°$, located at 32.45° E, 115.55° N. The upstream area of this point in GloFAS was almost the same as the actual area of the basin (the former was 31,364 km$^2$ and the latter was 30,630 km$^2$);
3. RHFS-S-ERA5 was produced by the RHFS forced with ERA5. This simulation, with the same hydrological forecast system as RHFS-S and the same input as GloFAS-S, was designed to study the effect of the input data as well as the modelling. The ERA5 basin–mean precipitation was calculated by applying IDW to the original grid precipitation of ERA5 and then used to force the RHFS system to produce RHFS-ERA5;
4. RHFS-S-QM was the result of applying quantile mapping to RHFS-S;
5. GloFAS-S-QM was the result of applying quantile mapping to GloFAS-S;
6. RHFS-S-ERA5-QM was the result of applying quantile mapping to RHFS-S-ERA5.

For RHFS-S-QM, GloFAS-S-QM and RHFS-S-ERA5-QM, QM training was conducted during the calibration period to estimate the values of the quantile–quantile relation between the natural discharge and each simulation.

Forecast

Determining the role of the initialization error and the model error was helpful in deciding which improvement aspect was more promising. We also paid attention to the influence of post-processing on the forecasts. To this end, six discharge reforecasts were analyzed, all forced by ECMWF reforecasts, with lead times between 1 and 7 days.

1. RHFS-F used meteorological observations as initialization of the RHFS, and the input was basin–mean precipitation reforecasts, which were converted from raw daily gridded ECMWF precipitation reforecasts by IDW. Due to the gaps in the observation time series in the winter half-year of the evaluated years, the reforecasts of the RHFS system were initialized separately from 1 June of each year. For example, when the start day of the reforecast was 2 October 2008, the initialization period was from 1 June 2008 to 1 October 2008. This meant that the initialization period was different for each reforecast depending on the time of year, the period being shortest in June, and longest in October. Although this had some impact on the simulations, overall, we believe it did not alter the results;
2. GloFAS-F is short for GloFAS reforecast and was the existing CEMS river discharge reforecast dataset from GloFAS, initialized with ERA5 and forced with ECMWF-ENS reforecasts [22]. GloFAS-F was downloaded for the same river pixel as GloFAS-S;
3. RHFS-F-ERA5 had the same configuration as RHFS-F, but used ERA5 as initialization data; it is used to evaluate the impact of the model error and the initialization error;
4. RHFS-F-QM was the result of applying quantile mapping to RHFS-F;
5. GloFAS-F-QM was the result of applying quantile mapping to GloFAS-F;
6. RHFS-F-ERA5-QM was the result of applying quantile mapping to RHFS-F-ERA5.

For RHFS-F-QM, GloFAS-F-QM and RHFS-F-ERA5-QM, the quantile mapping training was conducted during the calibration period to estimate the values of the quantile–quantile relation between the natural discharge and each raw forecast.

### 2.3.3. Quantile Mapping

Several studies have suggested that post-processing of discharge forecasts is effective in improving forecast quality [2]. In this paper, the quantile mapping (QM) method was used for statistical postprocessing, based on matching the cumulative distribution function (CDF) of the modelled time series to the CDF of observations. The effectiveness of QM depends on the CDF of the observations and the CDF of the modelled data having the same bias during the training and validation periods.

Several QM methods have been developed using different CDF estimating algorithms. In this paper, an empirical quantile called RQUANT was used to calculate the empirical CDF [35] from the R package "qmap". RQUANT uses local linear least squares regression to produce a robust estimate of the empirical quantile–quantile plot for the sequential quantiles, resulting in a table with empirical quantiles of simulation and a corresponding table with robust estimates of the empirical quantiles of observation. For any value outside of the post-processed range, the transformation is extrapolated using the slope of the local linear least squares regression at the outer most points.

### 2.3.4. Verification Metrics

The modified Kling–Gupta efficiency coefficient (KGE) and its three components were used to verify the quality of the discharge simulation and the ensemble mean discharge forecast. The continuous ranked probability score (CRPS) and its skill version (CRPSS), using a monthly observation sample of the climatology as the reference, were used as verification metrics of the ensemble discharge forecast. The relative flood peak error and the relative flood volume error were used to evaluate the quality of the flood event

simulations. Categorical verification scores in the form of probability of detection (POD), false alarm ratio (FAR), and equitable threat score (ETS) were used to evaluate the detection capacity of ERA5 regarding daily precipitation events. A detailed description of the metrics calculation can be found in Appendix A.

**3. Result**

We divided the analysis into four steps, with the evaluation of ERA5 precipitation data performed first, followed by the verification of the discharge simulations and the precipitation reforecast statistics, and finally the discharge reforecast results. The metrics calculation was performed for the verification period (2007, 2008, June–October).

*3.1. ERA5 Precipitation*

The modified Kling–Gupta efficiency coefficient (KGE) and its three components were calculated to compare ERA5 basin precipitation data with basin precipitation observations (Figure 3). The results showed a strikingly low correlation between the daily ERA5 precipitation data and the observed precipitation (R was about 0.4), which was the main reason for the low KGE. The bias ratio and variability ratio were relatively good, with the former nearly 1 and the latter 0.8.

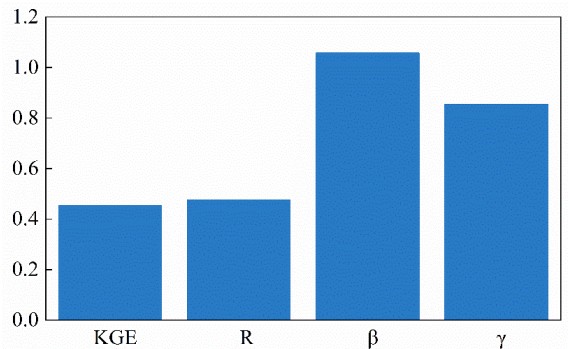

**Figure 3.** KGE, R, β, γ of the ERA5 basin precipitation data over the verification period (2007, 2008, June–October). KGE = modified Kling–Gupta efficiency coefficient; R = correlation; β = bias errors; γ = variability error.

Figure 4a shows the mean values of the ERA5 data and the basin precipitation observations for no rain (<0.1 mm/d), light precipitation (0.1–10 mm/d), moderate precipitation (10–25 mm/d) and heavy precipitation (25–50 mm/d) during the verification period; and Figure 4b shows the categorical verification scores (POD, probability of detection; FAR, false alarm ratio; and ETS, equitable threat score, used to evaluate the detection capacity) for the four categories. According to Figure 4a, ERA5 tended to overestimate no rain and light precipitation events slightly, and mildly underestimate moderate precipitation events. This was consistent with the finding of previous studies [36]. For heavy precipitation events, the difference between ERA5's and the observations' mean values was very obvious, with the former being 13 mm/d and the latter 44 mm/d. The detection capacity of ERA5 regarding daily precipitation events was not very good, as the ETSs were low (below 0.2 for no rain and below 0.1 for other precipitation events). Moreover, it was more difficult for ERA5 to accurately detect moderate and heavy daily precipitation events (for rainfall above 10 mm/d, PODs were below 0.2 and FARs about 0.8) than no rain and light precipitation events (PODs were above 0.4 and FARs below 0.6).

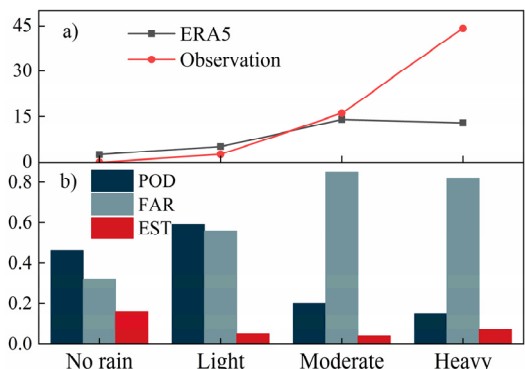

**Figure 4.** The (**a**) mean values of ERA5 data and of basin precipitation observations (mm/d), and (**b**) POD, FAR, and EST of ERA5, for four observed rain categories over the verification period (2007, 2008, Jun–Oct). POD = probability of detection; FAR = false alarm ratio; EST = equitable threat score; the four categories are: no rain (<0.1 mm/d), light precipitation (0.1–10 mm/d), moderate precipitation (10–25 mm/d), and heavy precipitation (25–50 mm/d).

The information in Figure 4 confirms the low KGE scores and shows that even though the precipitation values were correct on average (good bias), the system was not able to accurately predict when the event was expected to occur (poor timing shown by low correlation, as well as, to a lesser extent, by a variability of below 1, showing not enough day-to-day changes in ERA5).

*3.2. Discharge Simulation*

Figure 5 shows the time series of the six discharge simulations during the verification period (2007, 2008, June–October), together with the figures for the KGE and its three components. In general, RHFS-S, as expected, presented a better fit with the best KGE (0.88), followed by the GloFAS-S reanalysis (KGE of 0.54) and RHFS-S-ERA5 (KGE of 0.49), suggesting that the error introduced by using ERA5 data for forcing was the main reason for the poor simulation of the global hydrological forecast system. Additionally, RHFS-S-ERA5 had the worst bias ratio (0.52) among the raw simulations and an almost perfect variability ratio (1.03), suggesting that the use of ERA5 mainly resulted in the underestimation of the discharge. ERA5's slight overestimation of precipitation for no rain and light precipitation events had little impact on the discharge simulation, with the light blue line (RHFS-S) almost always above the light yellow line (RHFS-S-ERA5). However, its underestimation of moderate and heavy precipitation events led to the underestimation of flood peaks, except for the last peak in 2008. In contrast, GloFAS-S (light red line), when compared with the natural discharge (dashed black line) and RHFS-S-ERA5 (light yellow line), overestimated the baseflow, which contributed to its high bias ratio (1.26). The KGE performance of GloFAS was consistent with the results of Harrigan [22], that is, the bias was greater than 1 and the variability was less than 1 in China.

Quantile mapping is an effective way to partially eliminate errors originating from both the modelling system (uncalibrated global hydrological model) and the forcing data (ERA5), as the KGE after quantile mapping reached 0.74 (RHFS-S-ERA5-QM) and 0.84 (GloFAS-S-QM). However, QM could not improve the simulation performance of the locally calibrated regional hydrological forecast system forced with observations, leading to the KGE of RHFS-S-QM (0.8) being lower than the KGE of RHFS-S (0.88).

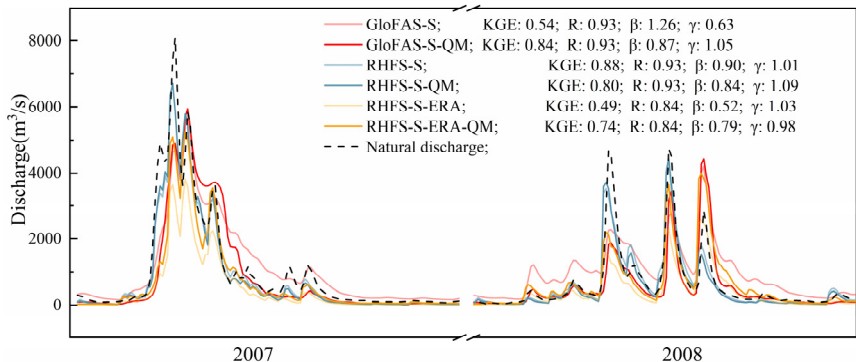

**Figure 5.** Time series of discharge simulations and the natural discharge for the verification period (2007, 2008, June–October), and associated KGE, R, β, γ. KGE = modified Kling–Gupta efficiency coefficient; R = correlation; β = bias errors; γ = variability error.

Table 2 shows the relative flood peak errors (ratio of simulated peak error to observations) for the four flood events (including one multi-peak flood event with four continuous peaks in 2007, and the three events in 2008) of the verification period, while Table 3 shows the relative flood volume errors (ratio of simulated volume error to observations), calculated excluding the baseflow. In general, and in most instances, the simulations underestimated the flood peak and flood volume. Among the raw simulations, RHFS had the smallest relative peak error and the smallest relative volume error. Regarding the peak simulations, on the one hand, GloFAS-S and RHFS-S-ERA5 had a larger relative error than RHFS-S (mean error of 36.2%, 50.9% and 23.9%, respectively), caused by the poor accuracy of ERA5, which failed to provide good simulations of moderate and heavy rain events. On the other hand, the GloFAS system could adapt well to the error characteristics of ERA5, as suggested by GloFAS-S having lower peak simulation errors than RHFS-S-ERA5. After excluding the influence of the baseflow, the flood volume errors were mainly attributable to the input, with a relatively small influence of the modelling system, resulting in absolute means of relative flood volume errors of 72.1% (GloFAS-S), 26.1% (RHFS-S) and 58.9% (RHFS-S-ERA5).

**Table 2.** Relative flood peak error of simulations for four flood events during the verification period.

| Flood Code | Relative Flood Peak Error (%) | | | | | |
|---|---|---|---|---|---|---|
| | GloFAS–S | GloFAS–S–QM | RHFS–S | RHFS–S–QM | RHFS–S–ERA | RHFS–S–ERA–QM |
| 1(1) | −57.8 | −68.7 | −30.7 | −26.3 | −80.7 | −72.8 |
| 1(2) | −42.2 | −39.5 | −27.6 | −27.5 | −60.7 | −42.8 |
| 1(3) | −1.8 | 1.8 | −7.0 | −6.8 | −38.5 | −13.6 |
| 1(4) | −10.2 | 3.2 | −22.5 | −21.8 | −44.1 | −13.6 |
| 2 | −51.3 | −60.9 | −29.7 | −24.7 | −67.7 | −54.9 |
| 3 | −43.6 | −33.2 | −7.0 | −6.1 | −50.6 | −22.6 |
| 4 | 46.7 | 54.4 | −42.7 | −53.1 | −13.8 | 33.7 |
| Absolute mean | 36.2 | 37.4 | 23.9 | 23.8 | 50.9 | 36.3 |

The improvement of the flood simulation by quantile mapping depended on the consistency of the error pattern during the training and the validation periods. Table 2 shows that QM was trained to increase the flood peak and the flood volume of RHFS-S-ERA. This resulted in the post-processing greatly reducing the errors in most cases (flood events 1–3), when RHFS-S-ERA underestimated the flood peak and volume, leading to an improvement of the average flood peak error from 50.9% to 36.3%, and a reduction in the flood volume errors for flood events 1–3 from 10% to 20%. There was a small change in the RHFS-S after QM, with changes in the relative flood peak and flood volume errors of 0.1% and 0.7%, respectively. For GloFAS-S, QM had a very random effect, reducing

or amplifying the flood peak, but the flood volume was always increased (the relative flood volume error was reduced by about 15% for flood events 1–3). Due to QM reducing the baseflow, as mentioned above, the flood volume excluding the baseflow was larger in relative terms.

**Table 3.** Relative flood volume error of simulations for four flood events during the verification period.

| Flood Code | Relative Flood Volume Error (%) | | | | | |
|---|---|---|---|---|---|---|
| | GloFAS−S | GloFAS−S−QM | RHFS−S | RHFS−S−QM | RHFS−S−ERA | RHFS−S−ERA−QM |
| 1 | −44.5 | −26.0 | −15.4 | −12.2 | −59 | −40.9 |
| 2 | −81.9 | −71.2 | −32.6 | −26.9 | −67.8 | −57.6 |
| 3 | −51.2 | −38.9 | 4.4 | 1.4 | −45.4 | −22.0 |
| 4 | 110.8 | 154.1 | −51.9 | −66.5 | 63.2 | 140.8 |
| Absolute mean | 72.1 | 72.6 | 26.1 | 26.8 | 58.9 | 65.3 |

### 3.3. Precipitation Reforecast

We evaluated the skill of the precipitation reforecasts based on CRPSS values (measuring forecast skill compared with reference forecast) of the precipitation ensemble reforecasts (Figure 6) and on the KGE (with three components) of the precipitation ensemble mean reforecasts (Figure 7), for lead times of 1–7 d.

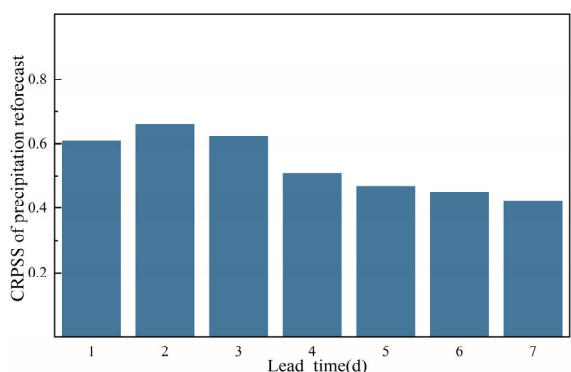

**Figure 6.** CRPSS (relative to sampled monthly mean climatology) of ECMWF basin–mean precipitation reforecasts at lead times of 1–7 d during the verification period (2007, 2008, June–October). CRPSS = continuous ranked probability skill score.

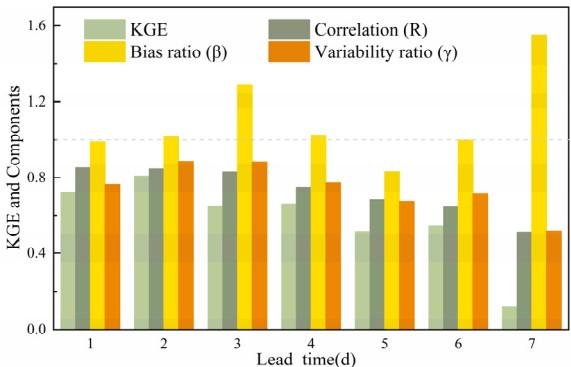

**Figure 7.** KGE, correlation (R), bias ratio (β) and variability ratio (γ) of the ECMWF precipitation reforecast ensemble mean over the verification period (2007, 2008, June–October). KGE = modified Kling–Gupta efficiency coefficient.

Figure 6 shows the CRPSS values plotted on a bar chart. The climatology was defined as the monthly mean of daily basin precipitation observations over the six flood seasons. The values were above 0, indicating that, over the full lead time considered, the precipitation ensemble reforecasts had more skill than the climatology. In general, the forecasting skill decreased with increasing lead times. However, the skill at a lead time of 1 d was lower than at a lead time of 2 d, which could be attributed to the effect of initialization.

Figure 7 shows the KGE and its three components for the precipitation ensemble reforecast mean. Similar to the CRPSS, the KGE, bias ratio and variability ratio all declined with increasing lead times. Among the three components, only the bias ratio had a value of above 1, with the others below 1. The low variability of the ensemble mean of the reforecasts was expected, as it has long been known that precipitation ensemble forecasts significantly underestimate heavy rain [28,29], hence also their ensemble means. The bias ratios of 1 or above, combined with an underestimation of heavy rain suggested by the low variability, were an indication of the ensemble mean systematically overestimating low rainfall or dry episodes.

*3.4. Discharge Reforecast*

3.4.1. Discharge Ensemble Forecast

Figure 8 shows the CRPSS of discharge ensemble reforecasts during the verification period for lead times of 1–7 d. The climatology was defined as the monthly mean of the daily natural discharge over the six flood seasons. Generally, RHFS-F had the best forecast skill amongst the three raw reforecasts, followed by RHFS-F-ERA5, with GloFAS-F coming last. The skill of the RHFS-F discharge reforecasts decreased with increasing lead times, similar to the skill of the precipitation ensemble reforecasts. However, the same was not true for the Glo FAS-S and RHFS-F-ERA5 reforecasts, because they were influenced by the combined impact of two or three interactive sources of non-negligible errors (initialization, model, input).

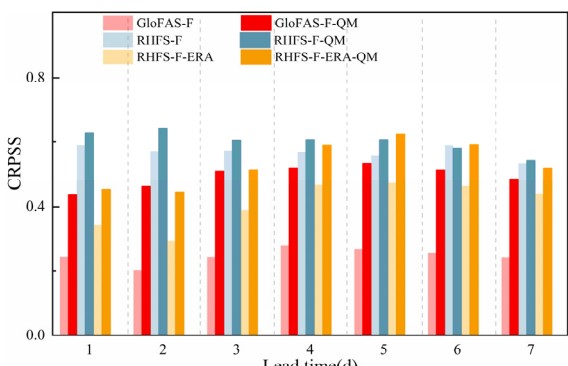

**Figure 8.** CRPSS (relative to sampled monthly mean climatology) of discharge reforecasts at lead times of 1–7 d during the verification period (2007, 2008, June–October). CRPSS = continuous ranked probability skill score.

The combined influence of initialization error and forecast model error on forecast skill became apparent by comparing the CRPSS of GloFAS-F with the CRPSS of RHFS-F; the impact of errors from the forecast system by comparing the CRPSS of GloFAS-F with the CRPSS of RHFS-F-ERA5; and the impact of the initialization error by comparing RHFS-F with RHFS-F-ERA5. Given that the CRPSS was generally affected by the model error and the low accuracy of initialization, it was below 0.3 for the GloFAS-F reforecasts, which was significantly lower than the CRPSS of RHFS-F (0.6). The errors from system, initial estimation and meteorological input were present at all lead times, with the influence of the initialization data dominating at shorter lead times, and was consistent with the research results of Zsoter et al. [37,38]. This was confirmed by the CRPSS difference between RHFS-F-ERA5 and RHFS-F, caused by different initializations (about 0.25), being much larger

than the CRPSS difference between RHFS-F-ERA5 and GloFAS-F, caused by the systems (about 0.1). At longer lead times, however, the model error primarily governed the forecast skill, which was demonstrated by the CRPSSs of RHFS-F-ERA5 and RHFS-F being close to each other and better than the CRPSS of GloFAS-F.

GloFAS-F-QM had the worst forecast skill, whereas RHFS-F-QM performed best at short lead times and had a similar CRPSS to RHFS-F-ERA5-QM at longer lead times. The difference between the CRPSS of GloFAS-F-QM and the CRPSS of RHFS-F-ERA5-QM was very small (dark red and dark yellow bar), which meant that QM was effective in reducing the model error. Additionally, the results suggested that QM also diminished the error from initialization, as shown by the CRPSS difference between RHFS-F and RHFS-F-ERA5 decreasing after QM, the former ranging between 0.1 and 0.3 and the latter between 0 and 0.2. The small improvement of RHFS-F after QM showed that QM also improved the error from input. In total, these examples demonstrated that post-processing is very necessary for ensemble discharge forecasting: it eliminates most of the errors caused by the system, as well as, to some extent, errors caused by the initial data and, to a minor extent, errors caused by meteorological input.

We then refined the analysis by subdividing the flow into low, moderate and high flow, defined by the 50th and 90th percentiles of daily natural discharges in the verification period (275 and 2730 m$^3$/s, respectively). Figure 9 shows the average CRPS of the reforecasts at different discharge magnitudes for the analyzed lead times, measuring the quality of the probabilistic forecasts. As can be seen from the figure, the skill of the ensemble forecasts was closely related to the magnitude of discharge, with better skill for low flow (lower CRPS) and worse skill for high flow (higher CRPS). Amongst the raw forecasts, RHFS-F performed best at moderate and low flow, followed by RHFS-F-ERA5 and GloFAS-F, which performed significantly worse. For high flow, RHFS-F performed best at lead times of 1 and 2 d, but GloFAS-F performed slightly better at lead times beyond 3 d.

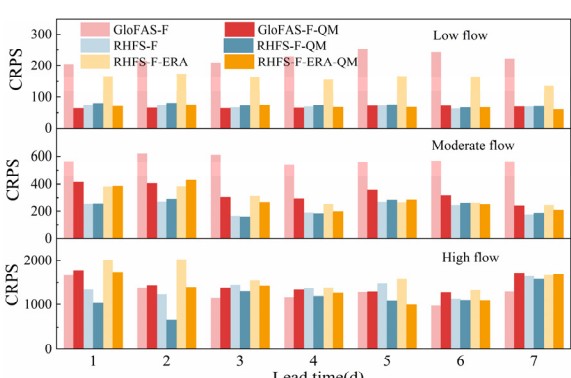

**Figure 9.** The mean CRPS of forecasts for low flow, moderate flow and high flow, respectively (defined by the 50th and 90th percentiles of the daily flow values), at lead times of 1–7 d, over the verification period (2007, 2008, June–October). CRPS = continuous ranked probability score.

The difference between RHFS-F and RHFS-F-ERA5 (light yellow and light blue bars, respectively) reflected the influence of the initialization error, which was present at low flow at all lead times (with small variations). However, for moderate flow, the difference decreased with increasing lead times and became very small at 5–7 d. For high flow, the difference could be ignored after 3 d. This illustrated that the impact of the initialization error on forecast performance was related to both the lead time and the flow magnitude. The longer the lead time and the higher the flow magnitude, the lower the impact.

A comparison between GloFAS-F and RHFS-F-ERA5 (light red and light yellow bars, respectively) showed the difference between the impact of the systems. With the same input and initialization, RHFS-F performed better in most cases, but GloFAS-F performed slightly better at high flow at longer lead times.

Quantile mapping improved the performance of RHFS-F and RHFS-F-ERA5 at different flow magnitudes. This was especially the case for the latter, where the most pronounced improvement was seen at low flow. However, the improvement of GloFAS-F after quantile mapping depended on the flow magnitude, with QM improving the prediction accuracy at moderate and low flow, but not significantly at high flow.

### 3.4.2. Discharge Ensemble Mean

Figure 10 highlights the KGE and its three components, calculated based on the ensemble mean of the discharge reforecasts for selected lead times. Surprisingly, the KGE of GloFAS-F-QM and of RHFS-F-ERA5-QM were both about 0.8 and appeared to be the best among all the discharge ensemble means we examined. In addition, the KGEs of GloFAS-F and of RHFS-F-ERA5 were below the KGE of RHFS-F, with the former two below 0.6 and the latter about 0.7.

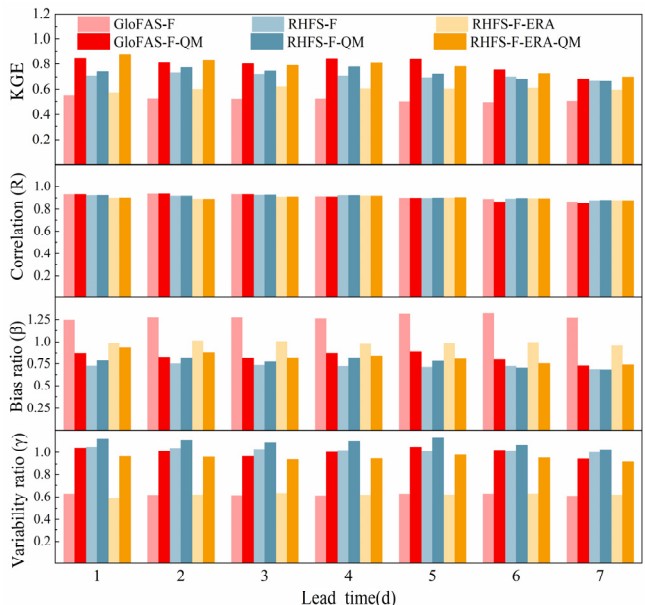

**Figure 10.** KGE, correlation (R), bias ratio (β) and variability ratio (γ) of the discharge reforecast ensemble mean at lead times of 1–7 d during the verification period (2007, 2008, June–October). KGE = modified Kling–Gupta efficiency coefficient.

In Figure 10, different lead times generated small variations only in the KGE and the three component scores of the daily discharge ensemble mean. When compared with the larger decrease in the precipitation ensemble mean (Figure 7), this indicated that the two hydrological prediction systems were relatively robust and could cope with the variable quality of the precipitation forecasts to produce discharge forecasts of stable quality. Because all forecasts have the same precipitation forecast input, the input error could not be tested in this study. For each of the six discharge reforecast datasets, the correlation between the ensemble mean and the natural discharge was almost the same (0.9) for all tested lead times. The bias ratio of the GloFAS-F reforecast ensemble mean was greater than 1.2, whereas the bias ratios of the other ensemble means were below 1, which was assumed to be the result of errors in the global system. The variability ratios of the experiments initialized with ERA5 (i.e., GloFAS-F and RHFS-F-ERA5) were about 0.6 for all lead times, which was much lower than the ratio for RHFS-F, which was initialized by observations. This indicated that the variability ratio was strongly affected by initialization errors for lead times of up to 7 d.

The quantile mapping had very little effect on the RHFS-F ensemble mean. However, QM increased the low variability ratio caused by errors from initialization, raising the γ of GloFAS-F and RHFS-F-ERA5 from 0.6 to about 0.9. Additionally, QM corrected the

excessive deviation of GloFAS-F, which was due to the poor forecast quality at low flow and mainly caused by model errors, lowering β from 1.3 to about 0.8.

## 4. Discussion and Conclusions

This paper explored the question of whether global hydrological forecast systems (GloFAS) are always inferior to regional hydrological forecast systems (RHFS), and examined the underlying factors (input, initialization and model). The main conclusions are summarized as follows:

1. Regarding river discharge simulations, GloFAS performed more poorly and was poorer than RHFS. This was mainly attributed to errors in the proxy-observations (ERA5) used as input to GloFAS, which did not provide good simulations of moderate and heavy daily precipitation events (above 10 mm/d), and to a model error which resulted in an overestimation of the baseflow. However, when the same ERA5 input was used, GloFAS appeared to be better than RHFS at simulating flood peaks;

2. On average, GloFAS showed a worse forecast performance than RHFS. This was mainly attributed to errors in the initial conditions (based on ERA5 initial data), and to model errors. However, for high flow forecasts, GloFAS was better than RHFS for longer lead times, and GloFAS was better for all lead times when RHFS was also initialized with ERA5;

3. Quantile mapping eliminated most of the initial errors, as well as part of the model and input errors, but failed to correct errors at high flow for GloFAS.

This study improves our understanding of the global hydrological forecast system in regional adaptability in comparison with the regional forecast system, and related influencing factors, thus, providing important guidance for basin discharge simulation and forecast, especially for flood ensemble forecast, and an improvement in the direction of the global hydrological forecast system. To the best of our knowledge, similar studies have not been conducted in the Chinese basin. Although this is an example of research on the upper Wangjiaba basin, the positive result may inspire research on the global hydrological forecast system with a focus on other basins. Our study could be used to supplement research on the existing regional hydrological forecast system, while on the other hand, the feedback of different basins to the global hydrological forecast system is also an important information source for improving the direction of ensemble forecasting, including ensemble forecasting models, data assimilation methods, post-processing methods, and so on. For a full test, GloFAS should have been forced with the observations for this catchment, but, due to gaps in the available observations, this was not technically possible. Improving the quality of ERA5 for moderate and higher precipitation could be an efficient way to improve GloFAS simulations and forecasts, given that GloFAS uses ERA5 as input to produce simulations and as initialization data to produce discharge forecasts. In addition, ERA5 precipitation data are closer to the observations over continental regions, such as Central Europe and the continental U.S., where many observations are used for assimilation in the production of ERA5 [39]. The higher ERA5 precipitation quality in these extratropical areas is also supported by the generally less common convective precipitation, and the difficulty of predicting convective precipitation causes ERA5 to show lower quality over tropical areas, especially during the summer season [40]. This means that, in regions where ERA5 has high precipitation quality, the global hydrological system may provide extra additional value over regional systems, as ERA5 is crucial in lowering the global system skill. We conclude that GloFAS represents a good alternative for flood ensemble forecasts for the Wangjiaba basin. Combining the low and medium flow forecast component of RHFS with the high flow flood forecasting component of GloFAS may also be an efficient way to generate ensemble discharge forecasts without needing to apply QM. In future studies, it would be beneficial to consider a longer period, covering more continuous years, to improve the robustness of results by increasing the sample size, and to run a longer simulation for initialization in a homogeneous way. In addition, according to Harrigan's research [22], the performance of GloFAS is related to the basin area, and the skill of increasing the basin size

is higher. Therefore, the simulation prediction results of GloFAS for large basins are worthy of attention.

**Author Contributions:** Data curation, H.W.; software, H.W. and E.Z.; visualization, H.W.; writing—original draft, H.W.; supervision, P.-a.Z. and F.P.; All authors contributed to conceptualization, formal analysis, investigation, methodology, validation, and writing—review and editing. All authors have read and agreed to the published version of the manuscript.

**Funding:** This research was funded by the National Natural Science Foundation of China (Grant No. 52009029, 52079037, 52279010). Han Wang was supported by a fellowship from the China Scholarship Council for her visit to the European Centre for Medium-Range Weather Forecasts (ECMWF), Reading, U.K. Ervin Zsoter was supported by the Wilkie Calvert Co-Supported PhD Studentships at the University of Reading. Christel Prudhomme and Ervin Zsoter were supported by the Copernicus Emergency Management Service—Early Warning Systems (CEMS-EWS).

**Data Availability Statement:** ERA5 data is available from the Copernicus Climate Change Service (C3S) Copernicus Climate Data Store (CDS), as well as GloFAS data and ECMWF reforecast.

**Acknowledgments:** We greatly thank the European Centre for Medium-Range Weather Forecasts, Reading, for providing the GloFAS data and precipitation data from ERA5.

**Conflicts of Interest:** The authors declare that they have no known competing financial interests or personal relationships that could have appeared to influence the work reported in this paper.

**Appendix A**

1.  The modified Kling–Gupta efficiency coefficient

The modified Kling–Gupta efficiency coefficient (KGE) [41] is of growing interest as a standard performance measure of hydraulics, which can be decomposed into three important components for assessing hydrological dynamics: correlation ($R$), bias error ($\beta$) and variability error ($\gamma$):

$$KGE' = 1 - \sqrt{(R-1)^2 + (\beta-1)^2 + (\gamma-1)^2} \tag{A1}$$

$$\beta = \frac{\mu_s}{\mu_o} \tag{A2}$$

$$\gamma = \frac{\sigma_s/\mu_s}{\sigma_o/\mu_o} \tag{A3}$$

where $R$ is the Pearson correlation coefficient between simulations (or ensemble mean forecasts, $s$) and observations ($o$); $\beta$ is the bias ratio; $\gamma$ is the variability ratio; $\mu, \sigma$ are the mean value and standard deviation of the variable, respectively. The KGE and its three decomposed components are all dimensionless with an optimum value of 1.

2.  Continuous ranked probability score

The continuous ranked probability score (CRPS) [27] compares the distribution of an ensemble forecast with the observed value. It is sensitive to bias in terms of forecast values as well as variability.

$$CRPS = \frac{1}{N} \sum_{i=1}^{N} \int_{-\infty}^{+\infty} [G_i(x) - H(x - o_i)] dx \tag{A4}$$

$$\begin{aligned} H(x - o_i) &= 1 \quad x \geq o_i \\ H(x - o_i) &= 0 \quad x < o_i \end{aligned} \tag{A5}$$

where $G_i(x)$ is the cumulative distribution function of forecasts on day $i$; $o_i$ is the observation of day $i$; and N indicates the number of forecasts. The smaller the CRPS value, the better, and the best value is 0.

3. Continuous probability ranking scores skill

The continuous ranked probability skill score (CRPSS) [27] uses reference forecasts to normalize the CRPS of the model forecast, measuring its forecasting skills compared with the reference forecast:

$$CRPSS = 1 - \frac{CRPS_{forecast}}{CRPS_{ref}} \tag{A6}$$

where $CRPS_{forecast}$ represents the CRPS value of the forecast; and $CRPS_{ref}$ is the CRPS value of the reference forecast. When the CRPSS of the forecast is equal to zero, the forecast skill is equal to that of the reference forecast. A CRPSS greater than 0 indicates a forecast that is more skillful than the reference forecast, whereas a CRPSS smaller than 0 indicates less skill than the reference forecast. A CRPSS approaching one indicates a perfect forecast. The reference forecast for any calendar day in this study has been defined as the monthly average of the observations over the six flood seasons in that month, regardless of which day of the month it is. For example, the reference forecast on 1 June (or any day of June) is the average of the natural discharge data (or precipitation) from the observations taken during the month of June (as we have six flood seasons, it is the average value of $30 \times 6$ values).

4. Relative flood peak error and relative flood volume error

The relative flood peak error and relative flood volume error are commonly used in flood assessment [42–44]:

$$\Delta PE = 100\% * (PE - PE_{obs}) / PE_{obs} \tag{A7}$$

$$\Delta PV = 100\% * (PV - PV_{obs}) / PV_{obs} \tag{A8}$$

where $\Delta PE$ is the relative flood peak error; $PE$ is the simulated flood peak; $PE_{obs}$ is the flood peak of natural discharge; $\Delta PV$ is the relative flood volume error; $PV$ is the calculated flood volume; and $PV_{obs}$ is the flood volume of natural discharge. The optimal value of both metrics is 0, which means an unbiased discharge simulation. The flood volume does not include the baseflow here (Figure A1), the detailed calculation of the flood volume can be found in Fan's paper [42,45].

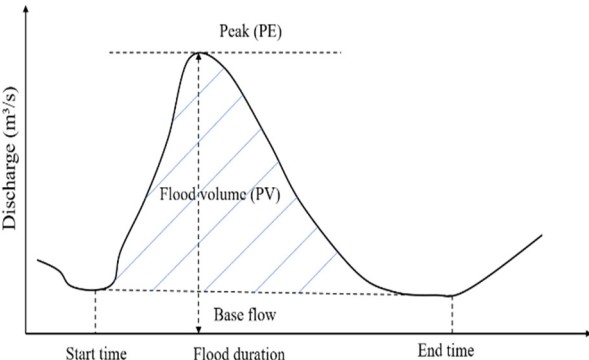

**Figure A1.** Typical flood hydrograph showing flood flow characteristics.

5. Probability of detection and false alarm ratio

The probability of detection POD and false alarm ratio (FAR) measure the probability of the correct detection and false detection by ERA5, respectively.

$$POD = \frac{hit}{hit + miss} \tag{A9}$$

$$FAR = \frac{false\ alarm}{hit + false\ alarm} \tag{A10}$$

A hit indicates that a simulated event occurred; a miss indicates that an event was not simulated to occur, but did occur; and a false alarm means that a simulated event did not occur. The optimal POD value is 1 and the optimal FAR value 0.

6.    Equitable threat score

The equitable threat score (ETS) represents a more comprehensive score than POD and FAR (applied here to examine ERA5's performance), because it penalizes both misses and false alarms in the same way [36].

$$ETS = \frac{hit - He}{hit + miss + false\ alarm - He} \tag{A11}$$

$$He = \frac{(hit + false\ alarm)(hit + miss)}{n} \tag{A12}$$

where *n* is the number of samples. The optimal ETS value is 1, the minimum is –1/3 and 0 indicates no skill.

**Appendix B**

The Xinanjiang model is a conceptual lump model, developed in 1973 and described in an international publication in 1980 [32]. It is widely used for humid and sub-humid regions in China. The structure of the Xinanjiang model is shown in Figure A2. The inputs of the model are basin–mean precipitation P, and basin–mean pan evaporation EM, while the outputs are basin outlet discharge Q and basin evaporation E. The calculation mainly consists of four parts: (1) calculation of evapotranspiration, with the evaporation coming from three soil layers, the upper, lower and deep soil layers; (2) runoff generation: this part assumes that runoff is produced by the basin when the soil moisture content reaches the field capacity; (3) runoff separation, using the free water storage reservoir for water source division, generating surface runoff, interflow and underground runoff; and (4) runoff routing module, divided into the river network confluence and the river channel confluence.

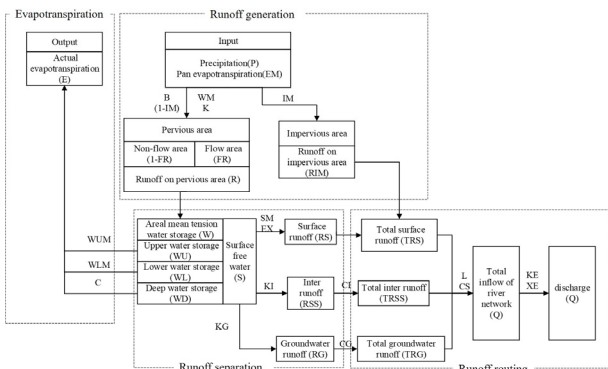

**Figure A2.** Structure of Xinanjiang model (adapted from [46]).

The parameters of the calibrated Xinanjiang model used in this paper are listed in Table A1, as well as the definitions and boundaries of the parameters.

**Table A1.** Definition of parameters, the boundaries and values of parameters used in this study.

| Parameter | Meaning | Boundary | Value |
|---|---|---|---|
| K | Ratio of potential evapotranspiration to pan evaporation | 0.1–1.5 | 1 |
| WUM | Upper layer soil water storage capacity | 5–30 | 24.8 |
| WLM | Lower layer soil water storage capacity | 60–90 | 78.2 |
| C | Deep evaporation coefficient | 0.09–0.3 | 0.1 |
| WM | Maximum watershed soil water storage capacity | 70–210 | 109.1 |
| B | Exponent of soil water storage capacity curve | 0.05–0.4 | 0.33 |
| IM | Percentage of impervious area in the catchment | 0–0.5 | 0.01 |
| SM | Free water storage capacity | 1–50 | 46 |
| EX | Exponent of soil water storage capacity curve | 1–1.5 | 1.4 |
| KG | Outflow coefficient of free water storage to groundwater | 0.2–0.6 | 0.4 |
| KI | Outflow coefficient of free water storage to interflow | 0.2–0.6 | 0.4 |
| CI | Recession constant of interflow | 0.1–0.99 | 0.78 |
| CG | Recession constant of groundwater runoff | 0.7–0.999 | 0.998 |
| CS | Recession constant of surface runoff | 0.01–0.4 | 0.32 |
| L | Lag in time | - | 1 |
| KE | Routing time in channel unit (d) | 0–1 | 1 |
| XE | Weight factor of Muskingum method | 0–0.5 | 0 |

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
