# Peer review of "Regional Adaptability of Global and Regional Hydrological Forecast System"

_water, doi:10.3390/w15020347_

Round 1

Reviewer 1 Report

1. Author must add results in abstract section (Numerical value) 2. More recent literature must be added in introduction section as well as in reference.  3. Flow chart/outline of manuscript must be added. 4. Please add the source of data in study area section. 5. Line 189- "Reginal" spelling mistake. 6. Uncertainty analysis of result must be added 7. Taylor diagram and Histogram must be added for better analysis 8. Limitation and advantages of methodology with rest to result must be added. 9. Compare your result with previous result must be included in result section.  10. A separate conclusion section is not written. 11. A clear description of future scope must be added. 

Reviewer 2 Report

Review of the manuscript Regional adaptability of global and regional hydrological fore 

cast system.

The authors compare two flood awareness system. The main questions were related to two issues “whether a global hydrological forecast system could be used as an alternative of regional system, or whether it could provide additional information”. The answers for those questions were obtained on the basis of results of the comparative analysis between two hydrological forecast system on the medium-range (30603km2) catchment located in humid climate condition. The results characterize the quality of two forecast systems in terms of discharge prediction and the source of the errors.

The article is well structured. The methodology is generally well presented. The figures are adequate in resolution and present the most important elements of the issues described in the article. The one elements, in reviewer opinion, should be improve. The final subchapter Discussion has a form of a conclusion section and only 2 positions of the literature is cited in this subsection. There is no scientific discussion related to the aspects covering in the main issues of the article. This element must be improved before publication.  

I recommend the publication of the paper after improvement of the discussion section.

Round 2

Reviewer 1 Report

Thank you for revising the manuscript.